# Identification of *MDK* as a Hypoxia- and Epithelial–Mesenchymal Transition-Related Gene Biomarker of Glioblastoma Based on a Novel Risk Model and In Vitro Experiments

**DOI:** 10.3390/biomedicines12010092

**Published:** 2024-01-01

**Authors:** Minqi Xia, Shiao Tong, Ling Gao

**Affiliations:** 1Department of Endocrinology & Metabolism, Renmin Hospital of Wuhan University, Wuhan 430060, China; 2Department of Neurosurgery, Renmin Hospital of Wuhan University, Wuhan 430060, China

**Keywords:** glioblastoma, risk model, EMT, hypoxia, immune, drug treatment

## Abstract

Background: Tumor cells are commonly exposed to a hypoxic environment, which can easily induce the epithelial–mesenchymal transition (EMT) of tumor cells, further affecting tumor proliferation, invasion, metastasis, and drug resistance. However, the predictive role of hypoxia and EMT-related genes in glioblastoma (GBM) has not been investigated. Methods: Intersection genes were identified by weighted correlation network analysis (WGCNA) and differential expression analyses, and a risk model was further constructed by LASSO and Cox analyses. Clinical, immune infiltration, tumor mutation, drug treatment, and enrichment profiles were analyzed based on the risk model. The expression level of the *MDK* gene was tested using RT-PCR, immunohistochemistry, and immunofluorescence. CCK8 and EdU were employed to determine the GBM cells’ capacity for proliferation while the migration and invasion ability were detected by a wound healing assay and transwell assay, respectively. Results: Based on the GBM data of the TCGA and GTEx databases, 58 intersection genes were identified, and a risk model was constructed. The model was verified in the CGGA cohort, and its accuracy was confirmed by the ROC curve (AUC = 0.807). After combining clinical subgroups, univariate and multivariate Cox regression analyses showed that risk score and age were independent risk factors for GBM patients. Furthermore, our subsequent analysis of immune infiltration, tumor mutation, and drug treatment showed that risk score and high- and low-risk groups were associated with multiple immune cells, mutated genes, and drugs. Enrichment analysis indicated that the differences between high- and low-risk groups were manifested in tumor-related pathways, including the PI3K-AKT and JAK-STAT pathways. Finally, in vivo experiments proved that the hypoxia environment promoted the expression of *MDK*, and *MDK* knockdown reduced the proliferation, migration, and EMT of GBM cells induced by hypoxia. Conclusions: Our novel prognostic correlation model provided more potential treatment strategies for GBM patients.

## 1. Introduction

Glioblastoma (GBM) is the most common primary malignant brain tumor in adults, with 3.2 cases per 100,000 people in the United States [1]. The total incidence of GBM has been growing year after year and will continue to rise as the population ages [2]. GBM, as an advanced glioma, has high invasiveness, a poor clinical prognosis, frequent recurrence, and a high mortality rate [3]. Patients with GBM, which accounts for 60% of gliomas, have a 5-year survival rate of only 4.7% [1]. Current treatment for GBM consists of maximal removal of the tumor followed by radiation and chemotherapy [4]. Despite recent progress in treatment strategies, GBM is prone to recurrence and has a near 100% fatality rate [5]. Therefore, a new diagnostic and therapeutic technique for GBM is urgently needed.

Hypoxia is an almost universal sign of malignant tumor growth, and tumor cells have a strong capacity to adapt in the face of the harsh environment of hypoxia. Studies have shown that the epithelial–mesenchymal transition (EMT) [6] of tumor cells can be induced by activating the hypoxia-inducible factors (HIFs) signaling pathway [7,8] in the face of hypoxia stress and further enhance the migration and invasive capacity of tumor cells and their adaptation to their surroundings. For example, melanoma, a highly advanced skin tumor, survives in the hypoxic environment and performs EMT by activating HIF-1α and blocking important immune system signaling pathways [9]. The accumulation of AKT in mitochondria can induce the phosphorylation of PDK1 at specific sites, reduce tumor cell death, maintain tumor cell proliferation, and induce EMT in the hypoxic environment [10]. The tumor environment is complex and variable, and the underlying mechanism of hypoxia-induced EMT in GBM patients has not been fully resolved. Targeted studies on GBM patients based on the characteristics of hypoxia-induced EMT are more comprehensive and feasible compared with separate studies on hypoxia or EMT in tumors.

Bioinformatics investigations of genome-wide RNA expression have revealed biological pathways that govern cell function and disease development at the molecular level [11,12], allowing for more accurate and personalized molecularly targeted therapeutics [13]. In this study, we used weighted gene co-expression network (WGCNA) and differential expression genes (DEGs) analyses to process the expression data of GBM in The Cancer Genome Atlas (TCGA) and Genotype-Tissue Expression (GTEx) databases. Then, based on least absolute shrinkage and selection operator (LASSO) and Cox analyses, we established a risk model consisting of the *MDK* and *STC1* genes, and we subjected this model to the analyses of clinical, immune, mutation, drug treatment, and functional enrichment. Finally, we selected *MDK* for further in vitro experiments. Our results suggest that the development of a differential expression risk score based on two genes (*STC1* and *MDK*) has potential value in predicting the prognosis and guiding the treatment of GBM patients.

## 2. Materials and Methods

### 2.1. Data Gathering

In this study, three public cohorts were gathered. Among them, the TCGA database (https://portal.gdc.cancer.gov/ (accessed on 10 October 2022), mRNA expression data and clinical details of GBM patients) [14] and GTEx database (https://www.gtexportal.org/home/index.html (accessed on 10 October 2022), mRNA information for normal brain tissues) [15] were used to establish a risk model for prognosis evaluation of GBM patients. The Chinese Glioma Genome Atlas (CGGA) database (http://www.cgga.org.cn (accessed on 10 October 2022), mRNA expression data and clinical details of GBM patients) [16,17] was used to further validate the risk model. Clinical information for all GBM patients is shown in Appendix A. In order to minimize any potential batch effects within or between the three cohorts, the “normalizeBetweenArrays” of the “limma” packages was used [18]. Moreover, we downloaded 863 hypoxia-related genes, 1253 EMT-related genes, and the c2.cp.kegg.v7.5.1.symbols dataset from the Molecular Signatures Database (MsigDB) (http://www.gsea-msigdb.org/gsea/downloads.jsp (accessed on 10 October 2022)) [19] for subsequent expression analysis and functional enrichment analysis. In addition, 300 chemokine genes and 149 immune checkpoint genes were obtained from the National Center for Biotechnology Information (NCBI) database (https://www.ncbi.nlm.nih.gov/ (accessed on 10 October 2022)) [20] for immune-related analysis.

### 2.2. Weighted Gene Co-Expression Network Analysis (WGCNA) 

The weighted gene co-expression network analysis (WGCNA) [21] was executed employing the R package “WGCNA”. Expression data from the TCGA and GTEx databases were subjected to clustering, utilizing an optimal soft power set to 14, leading to the construction of a dendrogram. Within the clustering results, genes exhibiting comparable characteristics were grouped into the same module.

### 2.3. Detection of Differentially Expressed Genes (DEGs)

The “limma” packages were employed for the identification of differentially expressed genes (DEGs) in the comparative analysis between GBM and normal brain samples. Subsequently, the ‘ggplot2’ package was utilized to visualize the DEGs through a volcano diagram. The criteria for defining DEGs were set at |logFC| > 1 and *p* < 0.001.

### 2.4. Risk Model Construction and Clinical Correlation Analysis

Firstly, by using the “Venn” package, intersection genes from the DEGs and WGCNA modules were obtained. The Venn diagram was drawn using the “Venn” package. Subsequently, prognosis-related genes were gained using univariate Cox analysis (*p* < 0.05) based on survival status, survival time, and expression levels of intersection genes in GBM patients, followed by LASSO regression analysis using the “glmnet” package to avoid overfitting. Furthermore, based on regression analysis and multivariate Cox analysis, the optimal risk model and coef values of genes in the model were obtained. We calculated a risk score for each patient according to the formula below:(1)$$Risk\;Score=\sum_{i=1}^{n}coef(i)*x(i)$$
where n is the number of prognosis-related genes, coef (i) is the regression coefficient, and x(i) is the gene expression level. The median risk score (1.0155–1.0161) of the TCGA cohort divided all GBM samples into high- and low-risk groups. Survival differences between high- and low-risk groups were calculated by the “survival” and “survminer” packages. Cox regression and Wilcoxon rank sum tests were used to evaluate the differences in survival probability and risk scores in clinical subgroups. The “survivalROC” package was used to calculate and visualize the AUC value to evaluate the accuracy of the risk model. Univariate and multivariate Cox analyses with clinical information were performed using the “survival” package. The “rms” package was utilized to develop a nomogram based on multivariate Cox regression coefficients.

### 2.5. Immune Infiltration Analysis

Firstly, the association between risk score and multiple immune cells was assessed using seven algorithms (XCELL, TIMER, QUANTISEQ, MCPCOUNTER, EPIC, CIBERSORT- abs, and CIBERSORT). Then, we used the “estimate”, “ggpubr”, and “limma” packages to compare the differences in the tumor microenvironment between the high- and low-risk groups, and we utilized Pearson correlation analyses to assess the association between risk score and DNA/RNAss. Following this, immune cell and functional differences between the high- and low-risk groups were further compared using single-sample gene set enrichment analysis (ssGSEA). Finally, the expression of the immune checkpoint and chemokine genes in the two groups was investigated by the “limma” package.

### 2.6. Mutation Analysis

Information on somatic mutations in GBM was downloaded from the TCGA database for 168 patients. Tumor mutation burden (TMB) was defined as the numbers of insertions/deletions and substitution mutations per million bases. Mutation in the high- and low-risk groups was analyzed by the “maftools” package. According to the median TMB score, all samples were divided into a high-TMB group and low-TMB group. Associations between risk score and TMB were shown in scatter plots, and differences in survival probability between the high- and low-TMB groups were analyzed by the Kaplan–Meier survival analysis.

### 2.7. Analysis of Drug Therapy

To ascertain the relationship between a drug and gene expression levels, the Genomics of Drug Sensitivity in Cancer (GDSC) (https://www.cancerrxgene.org/ (accessed on 10 October 2022)) [22] database and the Cancer Therapeutics Response Portal (CTRP) (https://portals.broadinstitute.org/ctrp/ (accessed on 10 October 2022)) [23] database were used. Furthermore, drug sensitivity and CCLE expression data from the PRISM database (https://depmap.org/repurposing (accessed on 10 October 2022)) [24] were collected to assess the relationship between drug sensitivity and risk score, as well as the differences in drug sensitivity between high- and low-risk groups.

### 2.8. Enrichment Analysis

Gene Ontology (GO), Kyoto Encyclopedia of Genes and Genomes (KEGG), and Gene Set Enrichment Analysis (GSEA) analyses used the “clusterProfile” package, and a *p*-value < 0.05 with FDR < 0.05 was regarded as statistically significant for the GO and KEGG enrichment analyses. In the GSEA enrichment analysis, the minimum to maximum number of genes was set at 15 to 500. After cyclic sampling 1000 times, the functional enrichment results of the high- and low-risk groups were obtained, and FDR < 0.25 was considered significant.

### 2.9. Immunohistochemistry (IHC)

The collected brain tissue paraffin sections were deparaffinized by xylene and dehydrated in ethanol. After blocking with 5% BSA, these sections reacted with anti-*MDK* (1:100, ZENBIO, R22526) overnight at 4 °C and reacted with the secondary antibody for 1.5 h. Then, these sections were visualized with a DAB kit (Servicebio, Wuhan, China) and captured using an Olympus fluorescent microscope.

### 2.10. Cell Culture and Transfection

GBM cell lines U251 and U87 were obtained from the Shanghai Institute of Biochemistry and Cell Biology (Shanghai, China). The cells were grown in High-Glucose DMEM containing 15% Fetal Bovine Serum (FBS, Gibco, CA, USA), 100 U/ML Penicillin, and 100 g/ML Streptomycin (Biosharp, CA, USA) at a suitable environment of 37 °C and 5% CO_2_. To specifically knock down the expression of *MDK*, the Si-NC and Si-*MDK* plasmids (sense: GGGAAGGGAAAGGACUAGA

TT; anti-sense: UCUAGUCCUUUCCCUUCCCTT) were purchased from RiboBio (Guangzhou, China) and were transfected to cells with a Lipofectamine3000 Kit (Invitrogen, Waltham, MA, USA) according to the guidance of the manufacturer.

### 2.11. RNA Extraction and Real-Time PCR

First, we collected all the RNA of GBM cells, and then these RNAs were combined as cDNA using the RevertAidTM cDNA Synthesis Kit (Thermo, Waltham, MA, USA). Finally, the SYBR^®^ Green Master Mix was utilized for RT-PCR to detect the total GAPDH and *MDK* levels according to the reagents’ guidance. All primers in this study were as follows: the forward primer of GAPDH, CCTTCCGTGTCCCCACT; the reverse primer of GAPDH, GCCTGCTTCACCACCTTC; the forward primer of *MDK*, CACCCCTAAGTGCCCAAA; the reverse primer of *MDK*, TGGGGAAGAACAAAAGCG.

### 2.12. Western Blotting (WB)

RIPA buffer (Servicebio, Wuhan, China) was utilized for cell lysis to obtain total cellular proteins, and 10 µg of these proteins were separated through SDS-PAGE electrophoresis according to different protein molecules and transferred to PVDF membranes. Next, these membranes were sealed with a quick blocking solution (Servicebio, Wuhan, China) and incubated with the corresponding primary antibodies and HRP-conjugated secondary antibodies in turn. Finally, the enhanced ECL Reagent (Biosharp, CA, USA) was utilized to expose and visualize the blots. The primary antibodies in this study included anti-*MDK* (1:1000, R22526, ZENBIO, Suzhou, China), anti-E-cadherin (1:1000, Proteintech, 20874-1-AP), anti-Vimentin (1:1000, 10366-1-AP, Proteintech, Waltham, MA, USA), anti-snail (1:1000, 13099-1-AP, Proteintech, Waltham, MA, USA), and anti-tubulin (1:5000, M20005, Absmart, Shenzhen, China).

### 2.13. Immunofluorescence Staining

The treated cells were immobilized for 15 min by paraformaldehyde and permeabilized for 10 min by 0.2% Triton. Next, these cells were blocked for 45 min by 5% BSA (Sigma, Shanghai, China) and incubated overnight with anti-*MDK* (1:100, R22526, ZENBIO) at 4 °C. These cells were exposed to Alexa Fluor 488-conjugated secondary antibody (1:100, GB25301, Servicebio) for 30 min and were then captured and analyzed with an Olympus fluorescent microscope.

### 2.14. Cell Viability and EdU Assay

A CCK8 kit (Biosharp, CA, USA) was utilized to detect the cell viability. The transfected cells were planted in 96-well plates with 5000 cells per well. After a period of culture, these cells were incubated with CCK8 reagent for 1 h. Finally, the absorbance (450 nm) was measured and analyzed using a microplate reader (Molecular Devices, CA, USA). For the EdU assay, after incubation with the EdU solution (Beyotime, Shanghai, China) in a dark environment at 37 °C for 1 h, the cells were immobilized with paraformaldehyde for 20 min, and DAPI (Sigma, Shanghai, China) was utilized to visualize the nucleus. Subsequently, EDU-positive cells were captured and counted using an Olympus fluorescent microscope.

### 2.15. Wound Healing and Transwell Assays

For wound healing, in short, the transfected cells were planted in 6-well plates and cultivated to the density of 80%. A pipette tip (100 µL) was used to create scratches and the cells were incubated with serum-free medium for 24 h. Finally, the wound healing photographs were captured using an inverted microscope (Olympus, Tokyo, Japan) and the movement distance of cells was calculated using Image J. For the transwell assay, the transfected cells were planted in the upper transwell chamber supplemented with the serum-free medium, and the serum medium was added into the lower transwell chamber. After incubation at 37 °C for 24 h, these cells were immobilized and stained. Subsequently, an inverted microscope (Olympus, Japan) was utilized to take images and count the cell numbers.

### 2.16. Statistical Analysis

R studio (version 4.2.0) and GraphPad Prism (9.0.0) software were used to analyze all the data. To analyze the differences between the two groups, Student’s *t*-test was used. For comparisons involving three or more groups, one-way analysis of variance (ANOVA) was employed. A statistically significant difference was defined as *p* < 0.05.

## 3. Results

### 3.1. The Flowchart of the Study

The research flow chart is shown in Figure 1. GBM expression data from the TCGA and GTEx cohorts were used to obtain 330 hypoxia-related DEGs, 607 EMT-related DEGs, and 2314 genes from the MEdarkgrey module, respectively. The intersection genes were further used for subsequent clinical, immune infiltration, mutation, drug treatment, and enrichment analysis. Furthermore, we selected the prognostic related gene *MDK* for further in vitro validation, including the expression of *MDK* in the environment of hypoxia, the proliferation and migration situation of GBM cells after *MDK* knockdown, and the EMT of GBM cells after *MDK* knockdown under a hypoxia condition.

### 3.2. Identification of DEGs in Collected Datasets

We combined the TCGA and GTEx cohorts to analyze the DEGs between GBM and healthy samples. We obtained 607 EMT-related DEGs (Figure 2A) and 330 hypoxia-related DEGs (Figure 2B) by setting |logFC| > 1 and *p* < 0.001 as the criteria. The TCGA and GTEx combined cohort was then used to create a co-expression network using the “WGCNA” package. Based on two evaluation criteria (scale independence and average connectivity), a soft threshold of 14 was set (Figure 2C,D) and genes were divided into 11 modules, with the MEdarkgrey module having the highest correlation with gene characteristics, comprising 2314 genes (Figure 2E,F). The 59 intersection genes in the three sets of filtered genes were then identified using Venn diagrams (Figure 2G).

### 3.3. Construction of Risk Models and Clinical Correlation Analysis

We used intersection genes to perform univariate Cox analysis, LASSO regression analysis, and multivariate Cox analysis in order to create the most efficient risk model for analyzing clinical traits and prognosis prediction. Screening by the criteria of *p* < 0.05, we obtained 22 genes that were connected to prognosis (Figure 3A). Further LASSO regression analysis and multivariate Cox regression analysis identified two genes (*STC1* and *MDK*) associated with prognosis (Figure 3B,C). Based on the level of the two genes and matching coefficients, the risk score was calculated: risk score = (0.2138 × *SEC1*) + (0.2702 × *MDK*). The TCGA-GBM cohort’s median risk score (1.0155 to 1.0161) was used to classify patients into two risk subgroups: high- and low-risk groups. The risk score, survival status, and two gene expression levels of GBM patients in the TCGA cohort are shown in Figure 3D. In addition, the Kaplan–Meier survival curve shows that all GBM patients in the high-risk group had significantly shorter survival probability than those in the low-risk group (Figure 3E). The results of the CGGA cohort’s risk score, survival status, two genes, and Kaplan–Meier survival curve were in line with those from the TCGA cohort (Figure 3F,G).

To investigate the relationship between the clinical subgroup and the probability of survival, the difference in the risk score between each clinical subgroup, we extracted the clinical data of age, gender, IDH status, 1p19q status, and MGMTp status, excluded samples with missing information, and used Cox regression analysis on the remaining samples, yielding the correlation between clinical subgroups and the probability of survival (Figure 4A–E) and the correlation between clinical subgroups and risk score (Figure 4F–J). Our results showed that patients aged ≥ 40 years had a significantly lower probability of survival compared to the patients aged < 40 years (*p* < 0.001). Patients with the IDH wild type have a significantly higher probability of survival than those with the IDH mutation (*p* < 0.05). In contrast, there appeared to be no significant differences in survival probability for gender (male and female), 1p19q status (codel and non-codel), and MGMTp status (unmethylated and methylated). In addition, we analyzed the differences in the risk score between subgroups. Age, IDH status, 1p19q status, and MGMTp status were different in the high- and low-risk groups. Risk scores were higher in the age ≥ 40 years, IDH wild type, 1p19q non-codel, and MGMTp non-methylation groups.

The ROC curve then confirmed that the risk score-based model had an AUC value of 0.807, indicating that it was reliable (Figure 5A). As shown in Figure 5B, a nomogram combining the risk score with other important clinical traits was constructed to predict patient survival at 1, 3, and 5 years in the TCGA cohort (C-index = 0.6410, 95% CI: 0.5911–0.6909, *p* = 3.1124 × 10^−8^). Furthermore, in both the univariate and multivariate Cox analyses, the *p* value of the risk score was less than 0.05, with HR > 1, indicating that the risk score was an independent risk factor for the prognosis of GBM patients (Figure 5C,D). In the CGGA validation cohort, the AUC value of the risk score-based model was 0.653 (Figure 5E). The C-index of the constructed nomogram was 0.6093, 95% CI: 0.5598–0.6588, *p* = 1.4823 × 10^−5^ (Figure 5F). Univariate and multivariate Cox analyses equally demonstrated that risk score was an independent prognostic factor for GBM patients (Figure 5G,H).

### 3.4. Overview of Immune Cell Infiltration

The relationship between the risk score and immune cells was examined. Firstly, we used seven algorithms to calculate the correlation between the risk score and multiple immune cells/functions. The results revealed that cancer-associated fibroblasts had a significant positive correlation with risk score (Figure 6A). As seen in Figure 6B, the Stromal score and Estimate Score for the high-risk group were significantly higher than in the low-risk group (*p* <0.05). Additionally, the relationship between the risk score and DNAss/RNAss is depicted in Figure 6C,D, and the findings reveal a significant negative relationship (R = −0.57, *p* < 0.001) between the risk score and RNAss. Finally, the expression of 16 immune cell types in the high- and low-risk groups were also examined using the ssGSEA algorithm. The box plot demonstrates that the proportion of immune cells were significantly higher in the high-risk group compared to the low-risk group (Figure 6E).

Immune checkpoints and chemokines are noted to be major factors of tumor immunotherapy because they regulate immune cell function. We looked into the variations in chemokines and immune system checkpoints between the high- and low-risk groups. The immune checkpoint-related genes, such as LIF, VEGFA, and SPP1, were found to be significantly elevated in the high-risk group (Figure 7A,B). And the chemokines’ related genes, for instance, CXCL8, CXCL2, THBS1, and POSTN, were elevated in the high-risk group when PI16 and ADCYAP1R1 were found to be significantly elevated in the low-risk group (Figure 7C,D).

### 3.5. Mutation Situation

We started with identifying the 15 most frequently mutated genes in the high- and low-risk groups (Figure 8A,B). Following that, TMB levels in the high- and low-risk groups were compared, and the relationship between risk score and TBM was investigated (Figure 8C,D). The findings revealed no discernible difference in TMB between the two groups and no significant relationship between the risk score and TMB score. According to the median TMB score, patients were split into two groups: high and low TMB score. Patients with high TMB had a higher probability of survival than those with low TMB, as shown in the Kaplan–Meier survival curve (Figure 8E). The worst prognosis was seen in the group of a low TMB score combined a high risk score (Figure 8F).

### 3.6. Drug Treatment

Using the GDSC and CTRP databases, we summarized the associations between gene expression and drug sensitivity in different cancers, with 30 drugs having the strongest correlations as listed (Figure 9A,B). We also analyzed the relationship between risk score and responsiveness to drugs (Figure 9C). The results showed that 20 drugs, LY2090314, TMC647055, TMC647055, methscopolamine, 3-AQC, niraparib, chloroxine, mCPP, L-732,138, BRD4770, AZD2858, nifekalant, simeprevir, hydrocortisone-hemisuccinate, CEP-32496, atiprimod, homochlorcyclizine, ciclesonid, AMG-232, CGM097, and idasanutlin, were positively associated with the risk score. Compared with the low-risk group, idasanutlin, CGM097, AMG-232, and CEP-32496 were significantly upregulated while LY2090314 and 3-AQC were significantly downregulated in the high-risk group (Figure 9D–I).

### 3.7. Results of Enrichment Analysis

We sought to elucidate the potential biological functions and pathways associated with the two prognostic genes (*STC1* and *MDK*). According to the expression of the high- and low-risk groups, we identified eight DEGs (|logFC| > 1 and *p* < 0.05). Then, KEGG enrichment analysis showed that the differences between high- and low-risk groups were mainly focused on cancer signaling pathways such as the PI3K-AKT and JAK-STAT pathways (Figure 10A). GO enrichment analysis showed that the differences between two groups were mainly related to the extracellular matrix (ECM) and cytokines (Figure 10B). Furthermore, GSEA enrichment analysis showed that the high-risk group was mainly enriched in cancer signaling pathways and the extracellular matrix while the low-risk group was mainly enriched in amino acid metabolism (Figure 10C).

### 3.8. MDK Is Elevated in GBM Patients and Related to Poor Prognosis

In previous studies, we found that hypoxia increased the expression of *MDK* in human umbilical vein endothelial cells [25] and *MDK* promoted the proliferation and migration of human breast cancer cells [26]. However, in GBM, the relationship between *MDK* and hypoxia and EMT has not been researched. Therefore, we chose *MDK* for further in vitro experiments to verify the efficacy of our model. First, in the TCGA database, we found that *MDK* was highly expressed in GBM (Figure 11A) and closely related to important clinical pathological features such as IDH wild type and 1p19q co-deletion (Figure 11B,C). At the same time, our clinical samples achieved the same results (Figure 11D,E). Finally, to verify the impact of *MDK* on the prognosis of GBM, the K-M curve demonstrated that the patients with high expression of *MDK* had a worse prognosis (Figure 11F–H). In conclusion, *MDK* was highly expressed in GBM and might be a potential prognostic marker.

### 3.9. Hypoxia Increases MDK Expression in GBM Cells

The growth of GBM is often accompanied by hypoxia, which causes GBM cells to alter the heredity and metabolism to adapt to this environment. In this article, we attempted to investigate the connection between *MDK* and hypoxia in GBM. First, in the TCGA database, we found that the expression of *MDK* was significantly positively correlated with hypoxia-related markers (HIF1A, HK2, VEGFA, and CA9) (Figure 12A). Then, GBM cells were cultured under 1% O_2_ conditions to simulate a hypoxia environment. According to PCR findings, hypoxia significantly increased the expression of *MDK* mRNA (Figure 12B). As shown in Figure 12C,D, *MDK* protein expression peaked after 24 h of hypoxia. At the same time, the fluorescence intensity of *MDK* in the hypoxia group was significantly increased (Figure 12E–H). In conclusion, hypoxia significantly induced the expression of *MDK* in GBM cells.

### 3.10. Knockdown MDK Inhibits GBM Cells’ Proliferation and EMT

To analyze the role of *MDK* on the proliferation and EMT of GBM cells, we knocked down the expression of *MDK* by transient transfection and WB was utilized to verify the success of plasmid construction (Figure 13G,H). As shown in Figure 13A–C, the knockdown of *MDK* significantly inhibited the proliferation of GBM cells. In the TCGA database, the results revealed that *MDK* was significantly positively correlated with EMT-related markers (ZEB1, Snail1, Vimentin, and N-cadherin) (Figure 13D). Additionally, the knockdown of *MDK* reduced the number of GBM cells invading (Figure 13E,F). According to WB results, the knockdown of *MDK* dramatically enhanced the expression of epithelial cell-related marker E-cadherin while significantly decreasing the expression of mesenchymal cell-related markers (Vimentin and Snail1) (Figure 13G,H). In short, the knockdown of *MDK* inhibited the proliferation and EMT of GBM cells.

### 3.11. Knockdown MDK Reverses Hypoxia-Induced EMT in GBM Cells

A hypoxic environment is regarded as the main driving force of the malignant behavior of tumors and stimulates EMT to enhance tumor metastasis and diffusion [27]. In our study, we found that hypoxia can induce the expression of *MDK* in GBM cells, and then we speculated whether *MDK* mediates hypoxia-induced EMT. As shown in Figure 14A–C, hypoxia significantly increased the migratory distance of GBM cells, but the knockdown of *MDK* reversed this result. In addition, the knockdown of *MDK* blocked the enhancement of the invasion of GBM cells by hypoxia (Figure 14D,E). Finally, the knockdown of *MDK* decreased hypoxia-induced Vimentin and Snaila expression and increased E-cadherin expression (Figure 14F,G). These results indicated that *MDK* mediates hypoxia-induced EMT.

## 4. Discussion

GBM has the characteristics of high incidence, strong invasiveness, low survival, and a high recurrence rate. Although the traditional treatment methods have made great progress, they are still not enough to solve the problem of frequent recurrence in GBM patients, which prompt people to more actively study the pathophysiology and molecular targeted therapy of gliomas. Hypoxia can induce EMT, promote tumor migration, and promote invasion, anti-apoptosis, and the degradation of the extracellular matrix. As far as we are aware, this is the first paper combining bioinformatics techniques to construct a risk model and investigate the EMT induced by hypoxia in GBM patients. In this paper, we used multiple public datasets to investigate the predictive role of novel biomarkers in GBM patients. The differential expression of GBM and normal samples and WGCNA analysis were used to identify 59 differentially expressed genes related to both hypoxia and EMT. LASSO and Cox regression analyses were used to screen out the best two prognostic genes (*STC1* and *MDK*) and construct the risk model. The risk score could accurately predict the overall survival of GBM patients and serve as an independent risk factor. Based on the risk model, we further discussed the differences in high- and low-risk groups with common clinical traits, immune cell infiltration, tumor mutation, drug treatment, and enrichment analysis. The prognostic gene *MDK* was used for an in vitro experiment to verify the promoting effect of *MDK* on tumor proliferation, invasion, and metastasis in the hypoxic environment. This investigation marks the initial discovery that the downregulation of the *MDK* gene effectively diminishes the proliferation, migration, and epithelial–mesenchymal transition (EMT) of glioblastoma multiforme (GBM) cells in response to hypoxic conditions.

In this study, we constructed a risk model that included two genes: *STC1* and *MDK*. *STC1* is a protein-coding gene associated with diseases including colon [28] and breast cancers [29], it can enhance the metastasis potential of HCC through the JNK signaling pathway [30], and it reduces the immune function of macrophages [31,32,33]. *MDK* is a secreted protein that acts as a cytokine and growth factor, and it is mediated by cell surface proteoglycan and non-proteoglycan receptors [34,35,36,37,38,39,40,41]. It regulates the inflammatory response, cell proliferation, cell adhesion, cell growth, cell survival, tissue regeneration, cell differentiation, cell migration, and other processes [35,36,37,38,39,40,41,42,43]. For example, *MDK* can promote T-cell proliferation through the NFAT signaling pathway and Th1 cell differentiation, inhibit the emergence of drug-resistant dendritic cells, and thereby inhibit the differentiation of regulatory T cells [43]. It has also been reported that the *MDK* gene can promote the proliferation of human cancer cells, such as gastric cancer cells [44] and glioma cells [45]. As a result of binding to anaplastic lymphoma kinase (ALK), insulin receptor substrate 1(IRS1) is phosphorylated and mitogen-activated protein kinase (MAPK) and pi3 kinase are activated, which promotes cell growth [40]. The transition of epithelial cells to mesenchymal cells is promoted through interaction with NOTCH2 [34]. Then, we performed a clinical analysis based on this two-gene risk model, and the Kaplan–Meier survival curve showed that the survival probability of patients in the high-risk group was significantly lower (*p* < 0.01). Patients with an age ≥ 40 years had a lower survival probability than those aged <40 years, and the IDH wild type group had a lower survival rate than the IDH mutation group. Clinical subgroup analyses showed that age ≥ 40, IDH wild type, 1p19q non-codel, and MGMTp un-methylated people have a higher risk score in the high-risk group compared to the low-risk group. Age has been reported as a risk factor for glioma [46], patients with IDH mutations have a better prognosis [47], patients with the 1p19q codel have a better prognosis [48,49], and MGMT methylated patients are more sensitive to treatment with TMZ [50,51]. Meanwhile, the AUC value of the risk score reached 0.807, giving our risk model an advantage in terms of accuracy compared to models constructed in previous studies [52,53]. In addition, the nomogram constructed with clinical information in the TCGA and CGGA cohorts could be used to predict patient survival at 1, 3, and 5 years, while univariate and multivariate Cox analyses showed that age and risk score were independent risk factors for patient prognosis.

In addition to prognostic value, we discovered that the risk model was linked to tumor immunity and tumor mutations. A growing body of evidence suggests that tumor immunity is important in tumorigenesis and treatment response [54,55]. The high-risk group had a significantly higher Stromal Score and Estimate Score than the low-risk group, according to immunological analysis. The risk score was significantly inversely related to RNAss, implying that as the risk score increased, tumor stem cell differentiation decreased. The proportion of immune cells in the high-risk group was significantly higher than in the low-risk group. As immune checkpoints or chemokines, LIF, VEGFA, SPP1, CCL2, CXCL8, THBS1, and POST genes were significantly higher in the high- and low-risk groups, suggesting that they could be used as therapeutic targets. For example, studies have shown that infiltrating tumor-associated macrophages promote tumor growth partly by secreting SPP1, which can promote glioma cell survival and angiogenesis [56]. In the presence of the Spp1 protein, KPDC cell migration was significantly increased, whereas basal KPC cell migration was significantly decreased after Spp1 inhibition [57]. VEGFA is also upregulated in many tumors, and its expression is associated with tumor development, including ovarian carcinomas, and is a target in many cancer therapies under development [58,59]. Tumor mutation analysis revealed that the missense mutations in the TTN gene were the most common, and the difference in TMB between the high- and low-risk groups was not statistically significant. However, patients in the high-TMB group had a lower survival probability with the lowest survival probability in the high-risk group in combination with high TMB. Those with higher TMB were more likely to be recognized and attacked by immune cells, and they responded more significantly to immunotherapy [60,61]. There were significant differences in drug sensitivity between the high- and low-risk groups for idasanutlin, CGM097, AMG-232, CEP-32496, LY2090314, and 3-AQC.

Furthermore, pathway enrichment revealed that the extracellular matrix and cytokine pathways were the most different between the high- and low-risk groups. The high-risk group was concentrated on cancer signaling pathways and the extracellular matrix, whereas the low-risk group was concentrated on amino acid metabolism. Cancer cells have been shown to regulate the cytokine environment in order to change the cell composition of the microenvironment, which promotes tumor progression [62,63,64]. Extracellular matrix (ECM) components in particular have been identified as key regulators of cancer progression [65].

Finally, we selected *MDK*, a prognostic gene, for a further in vitro experiment, which was shown to be highly expressed in GBM and associated with a poorer prognosis. The expression of *MDK* was elevated in the hypoxic environment, and the proliferation and migration ability of GBM cells were reduced after knockdown of the *MDK* gene. The knockdown of the *MDK* gene in the hypoxic environment could reverse hypoxia-induced EMT, indicating that the hypoxia-induced EMT of GBM is mediated by *MDK*.

In conclusion, we used bioinformatics to effectively construct a risk model which performed admirably in survival and prognosis prediction in GBM patients. The model’s predictions in training and validation cohorts verified its viability, and an in vitro experiment revealed that *MDK* mediates the EMT of tumor cells in a hypoxic environment. Nonetheless, our research has some limitations. On the one hand, our model is based on a retrospective analysis and will need to be validated in prospective studies. On the other hand, while this study is primarily based on bioinformatics techniques, the functional mechanisms and interactions of genes are important for validation, necessitating more complex and experimental data collection and evaluation.

## Figures and Tables

**Figure 1 biomedicines-12-00092-f001:**
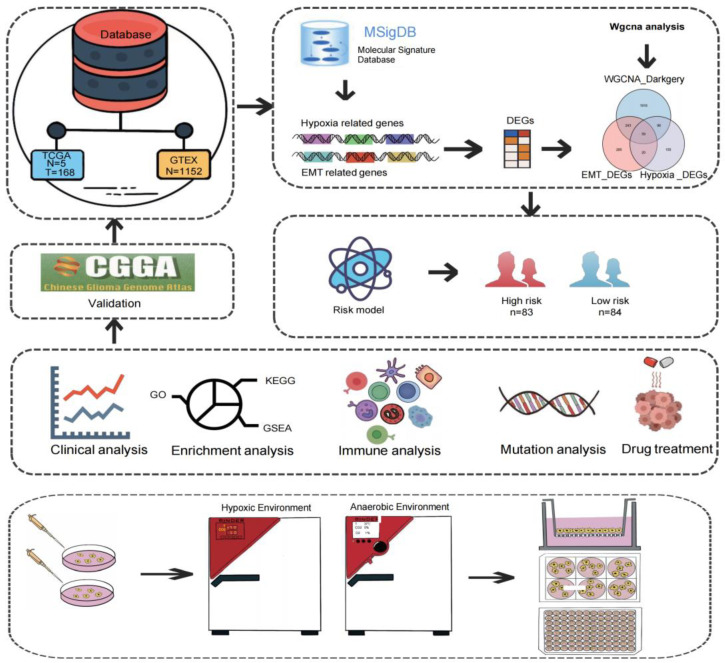
Flow chart of this study. Normal and tumor data of GBM patients in the TCGA and GTEx databases were used for WGCNA and differential analysis. The 59 genes obtained were further analyzed by Cox and LASSO regression analyses to construct a risk model. Based on the risk model, clinical correlation analysis, enrichment analysis, immune analysis, mutation analysis, and drug treatment analysis were continued. Finally, the *MDK* gene associated with prognosis was used for in vitro experiment verification.

**Figure 2 biomedicines-12-00092-f002:**
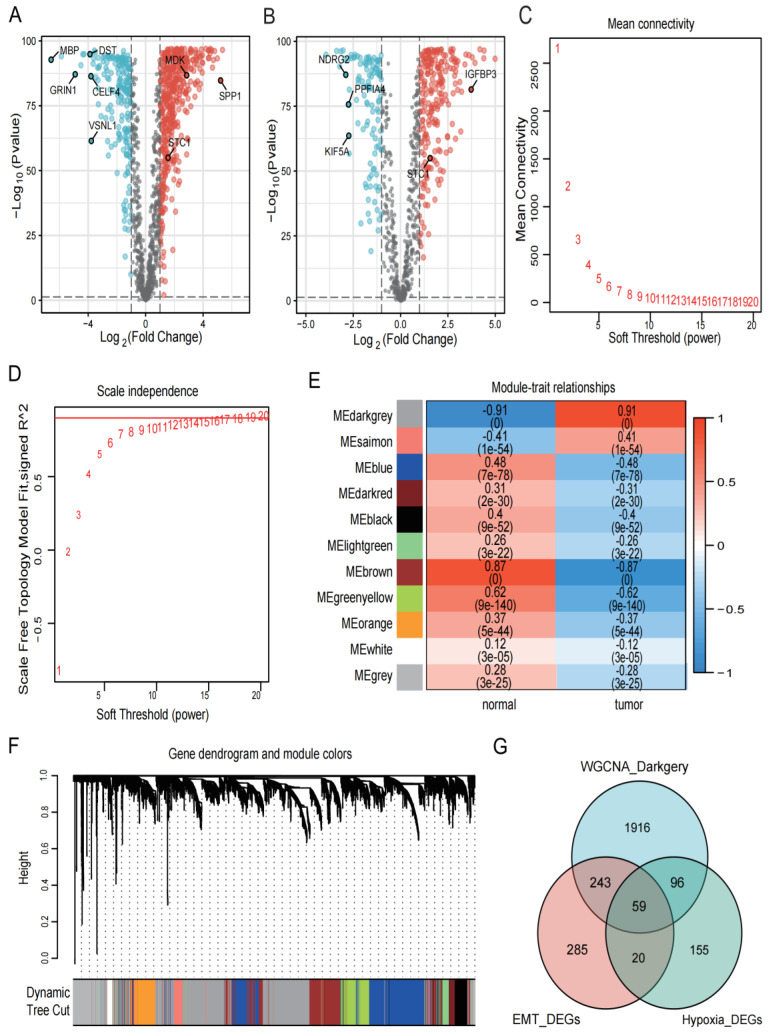
Identification of DEGs in the collected datasets. (**A**,**B**) Based on the TCGA and GTEx cohorts, EMT- and hypoxia-related DEGs were screened, respectively. Red represents upregulated genes with logFC > 1, black represents genes with no significant difference, and blue represents downregulated genes with logFC < −1. The *p* for both red and blue distributed genes was less than 0.001. (**C**,**D**) The selection of soft-thresholding power and construction of a scale-free network. (**E**) Module–trait relationships, where the ME darkgrey module has the highest correlation (cor = 0.91 and *p* = 0). (**F**) Cluster dendrogram in which similar genes are grouped into the same module. (**G**) Venn diagram screening 59 DEGs in 3 GBM cohorts.

**Figure 3 biomedicines-12-00092-f003:**
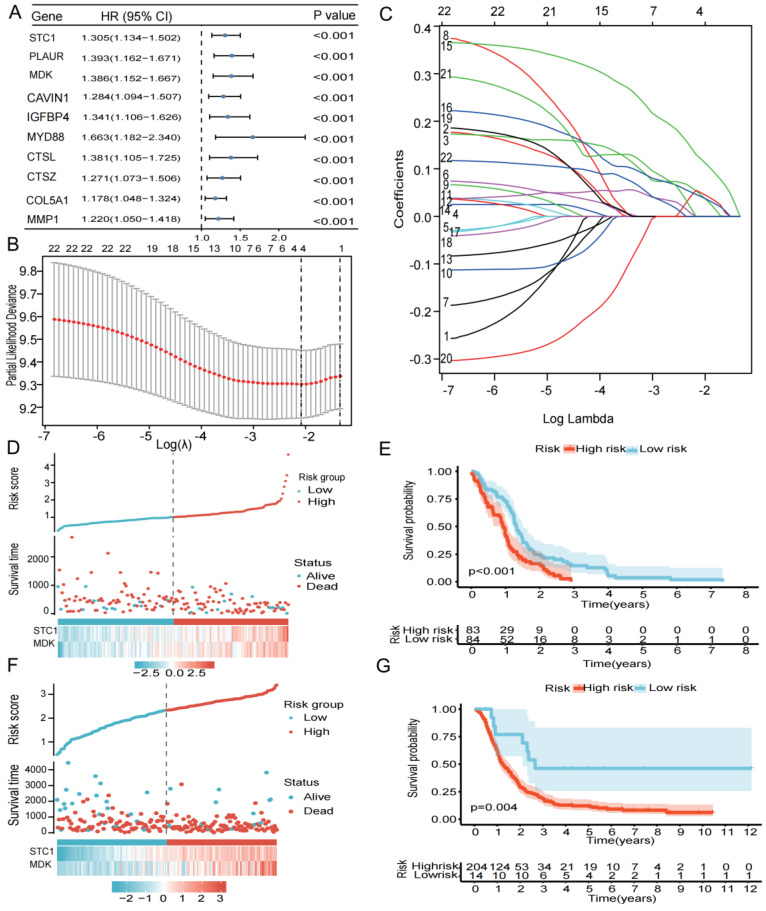
Construction and verification of a 17-gene risk signature. (**A**) Univariate Cox analysis of 59 DEGs. (**B**) LASSO coefficient profiles of the 59 DEGs. (**B**,**C**) Cross-validation for optimal parameter selection in the LASSO regression model. (**D**) The risk score, survival status, and risk genes in the TCGA cohort. (**E**) Kaplan–Meier survival analysis between the low- and high-risk groups in TCGA cohort, *p* < 0.05 was considered significant. (**F**) The risk score, survival status, and risk genes in the CGGA cohort. (**G**) Kaplan–Meier survival analysis between the low- and high-risk groups in CGGA validation cohort, *p* < 0.05 was considered significant.

**Figure 4 biomedicines-12-00092-f004:**
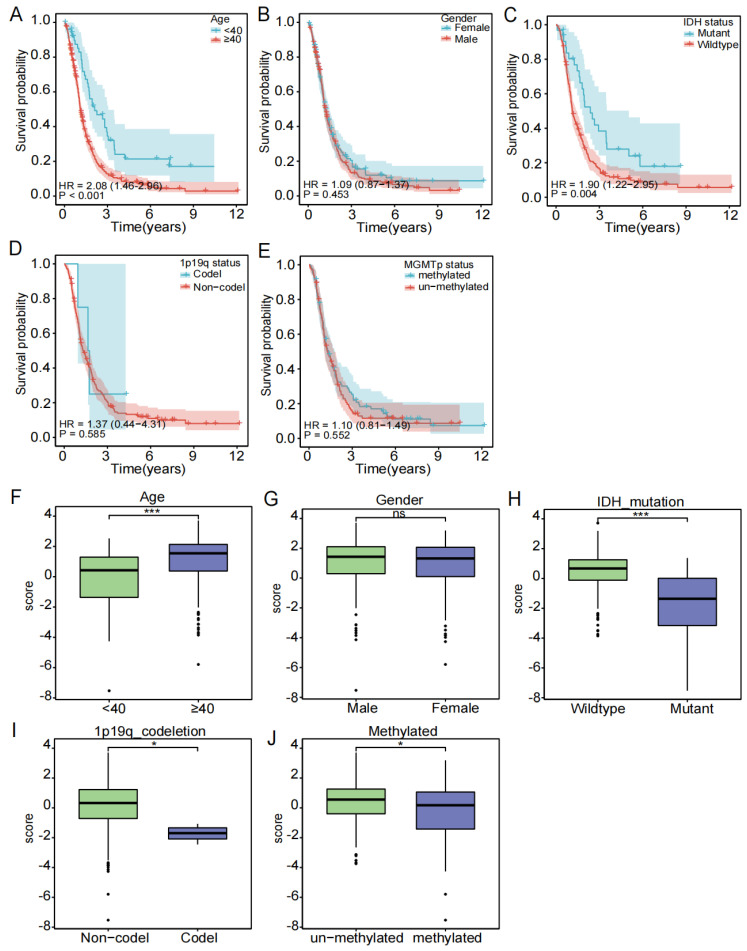
Subgroup clinical traits analysis. (**A**–**E**) Association of each clinical trait (age, gender, stage, TNM stage) with survival probability. (**F**–**J**) Association of each clinical trait (age, gender, stage, TNM stage) with risk score (ns is considered not significant, * *p* < 0.05, *** *p* < 0.001, *p* < 0.05 was considered significant).

**Figure 5 biomedicines-12-00092-f005:**
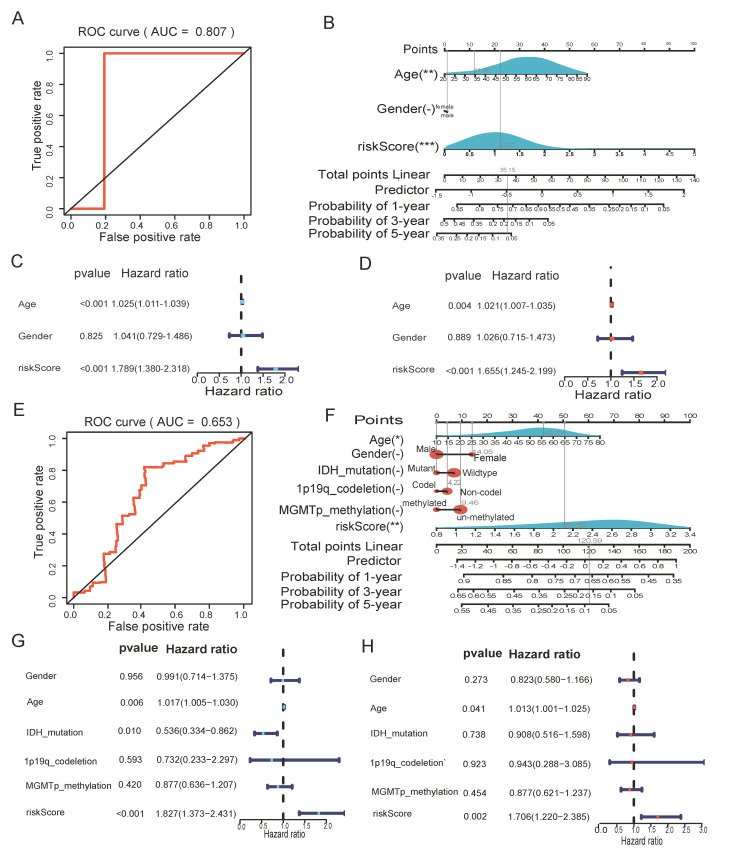
The risk score could be an independent factor for predicting the overall survival of GBM patients. (**A**) ROC curve for the risk model in the TCGA cohort. (**B**) Nomogram was based on clinical traits and risk score in CGGA validation cohort. (**C**) Univariate Cox analysis in the TCGA cohort. (**D**) Multivariate Cox analysis in the TCGA cohort. (**E**) ROC curve for the risk score in the CGGA validation cohort. (**F**) Nomogram was based on clinical traits and risk score in CGGA validation cohort. (**G**) Univariate Cox analysis in the CGGA validation cohort. (**H**) Multivariate Cox analysis in the CGGA validation cohort. (" – " is considered not significant, * *p* < 0.05, ** *p* < 0.01, *** *p* < 0.001, *p* <0.05 was considered significant).

**Figure 6 biomedicines-12-00092-f006:**
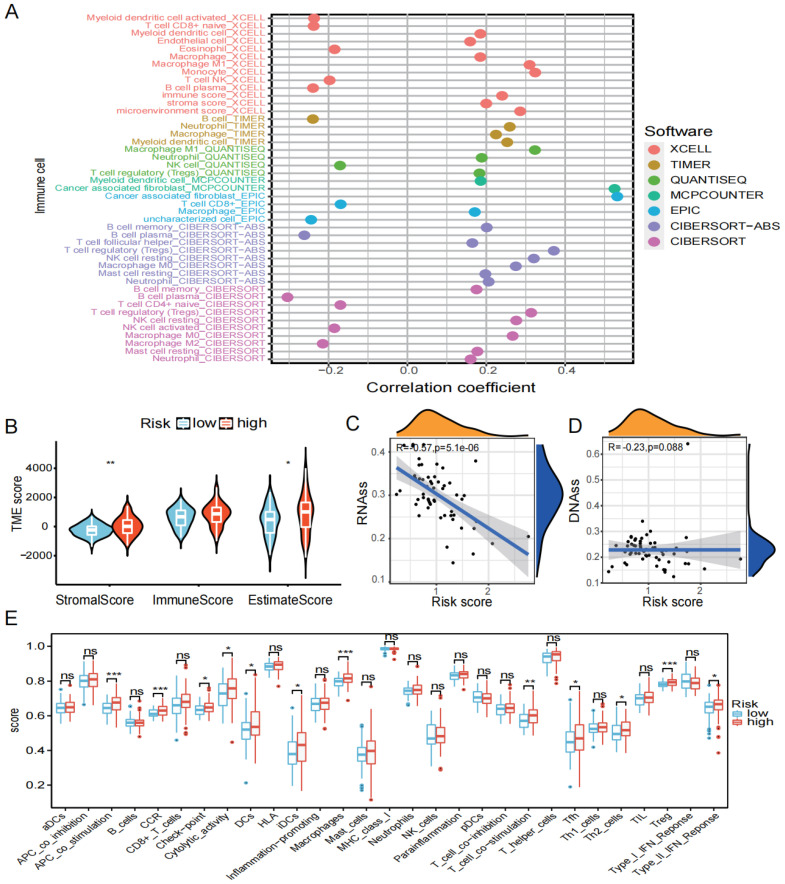
Immune infiltration analysis. (**A**) Seven algorithms (XCELL, TIMER, QUANTISEQ, MCPCOUNTER, EPIC, CIBERSORT-ABS, CIBERSORT) were used to determine the relationship between risk score and immune cells. (**B**) TME score for high- and low-risk groups. (**C**) Relationship between risk score and RNAss. (**D**) Relationship between risk score and DNAss. (**E**) Differences in immune cells and function between high- and low-risk groups (ns is considered not significant, * *p* < 0.05, ** *p* < 0.01, *** *p* < 0.001, *p* < 0.05 was considered significant).

**Figure 7 biomedicines-12-00092-f007:**
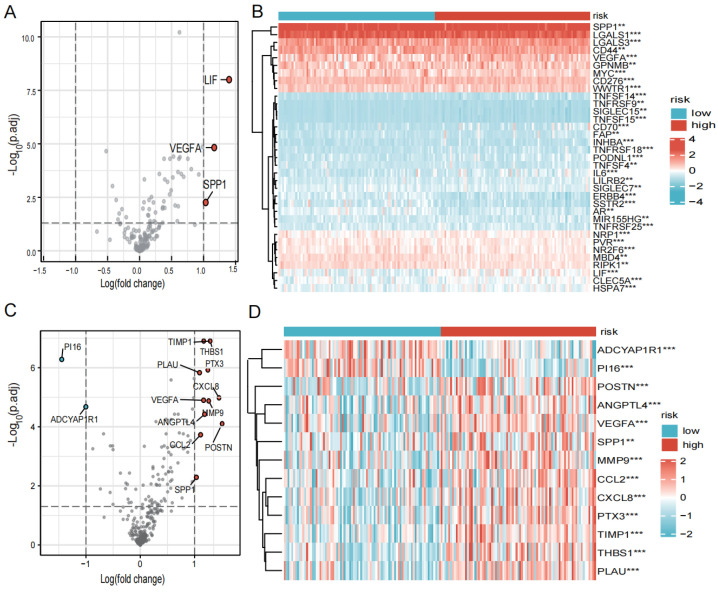
Expression of immune checkpoints, chemokines in high- and low-risk groups. (**A**) Volcano plot of the distribution of immune checkpoint-associated genes in high- and low-risk groups. The right side of the vertical axis is *p* < 0.05, logFC >1, and the left side is *p* < 0.05, logFC < −1. (**B**) Heatmap of the expression of immune checkpoint-related genes in high- and low-risk groups. (**C**) Volcano plot of the distribution of chemokines in high- and low-risk groups. The right side of the vertical axis is *p* < 0.05, logFC > 1, and the left side is *p* < 0.05, logFC < −1. (**D**) Heatmap of the expression of chemokines in high- and low-risk groups. (** *p* < 0.01, *** *p* < 0.001, *p* < 0.05 was considered significant).

**Figure 8 biomedicines-12-00092-f008:**
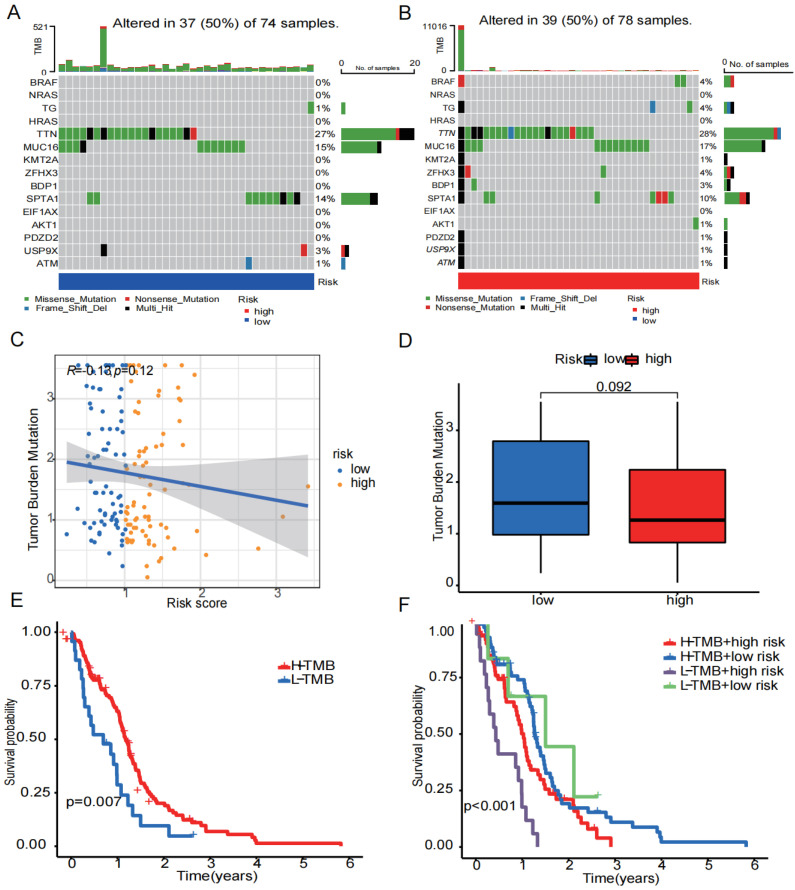
Tumor mutation status. (**A**) The 15 mutation genes in the high-risk group. (**B**) The 15 mutation genes in the low-risk group. (**C**) Relationship between risk scores and TMB. (**D**) Differences in TMB in high- and low-risk groups. (**E**) Differences in survival probability between the high- and low-TMB groups. (**F**) Differences in survival probability between high TMB in the low-risk group, high TMB in the high-risk group, low TMB in the low-risk group, and low TMB in the high-risk group. (*p* < 0.05 was considered significant).

**Figure 9 biomedicines-12-00092-f009:**
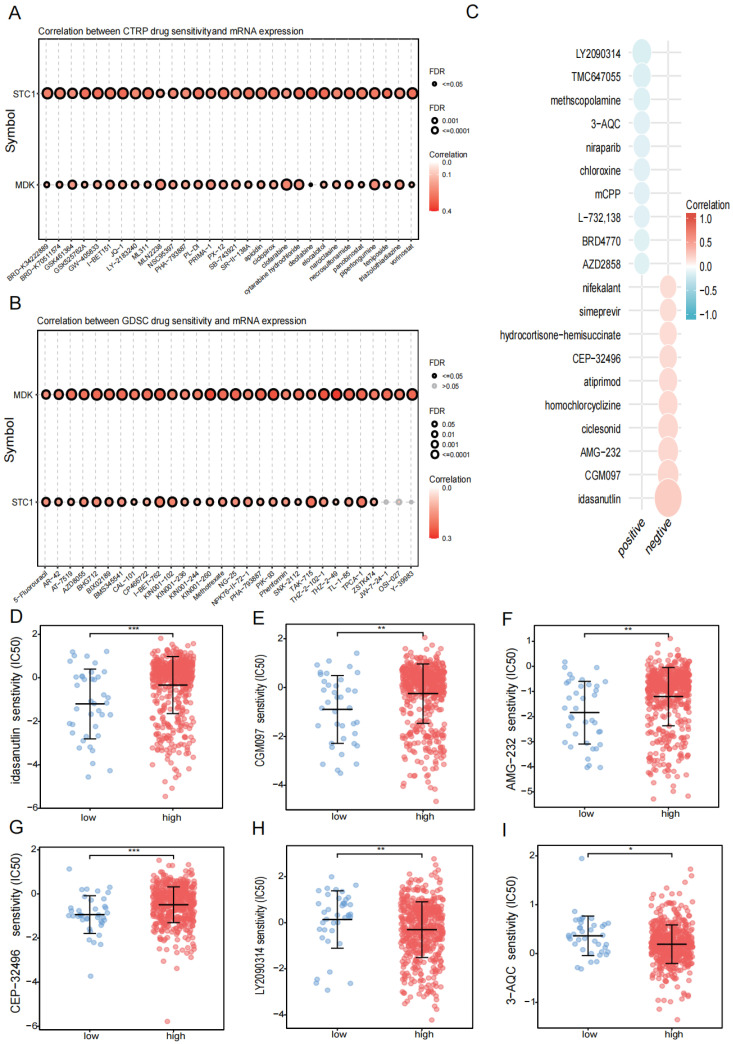
Drug treatment. (**A**) Relationship between risk score and drug sensitivity in the CTRP database. (**B**) The relationship between risk score and drug sensitivity in the GDSC database. (**C**) Relationship between risk scores and drug sensitivity in the PRISM database. (**D**–**I**) Relationship between risk scores and drug sensitivity in drugs (idasanutlin, CGM097, AMG-232, CEP-32496, LY2090314, 3-AQC). (* *p* < 0.05, ** *p* < 0.01, *** *p* < 0.001, *p* < 0.05 was considered significant.)

**Figure 10 biomedicines-12-00092-f010:**
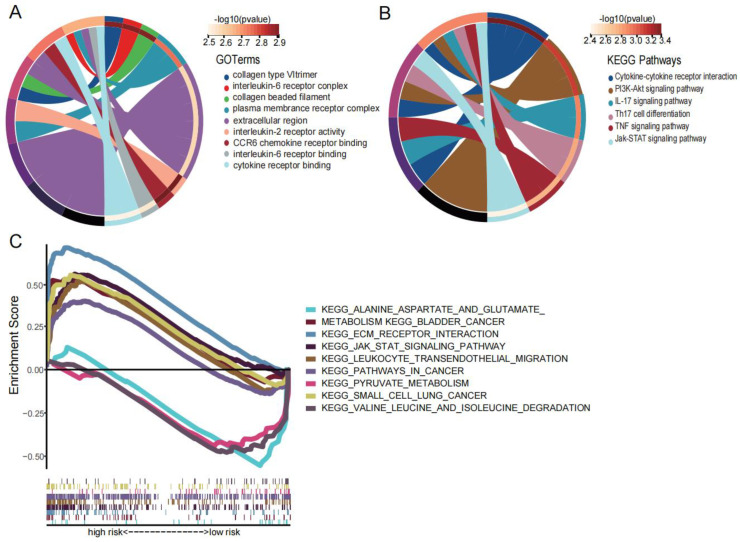
Enrichment analysis. (**A**) The KEGG diagram was made according to the DEGs between the high- and low-risk groups. (**B**) The GO diagram was made according to the DEGs between the high- and low-risk groups. (**C**) Gene set enrichment analysis of high- and low-risk groups.

**Figure 11 biomedicines-12-00092-f011:**
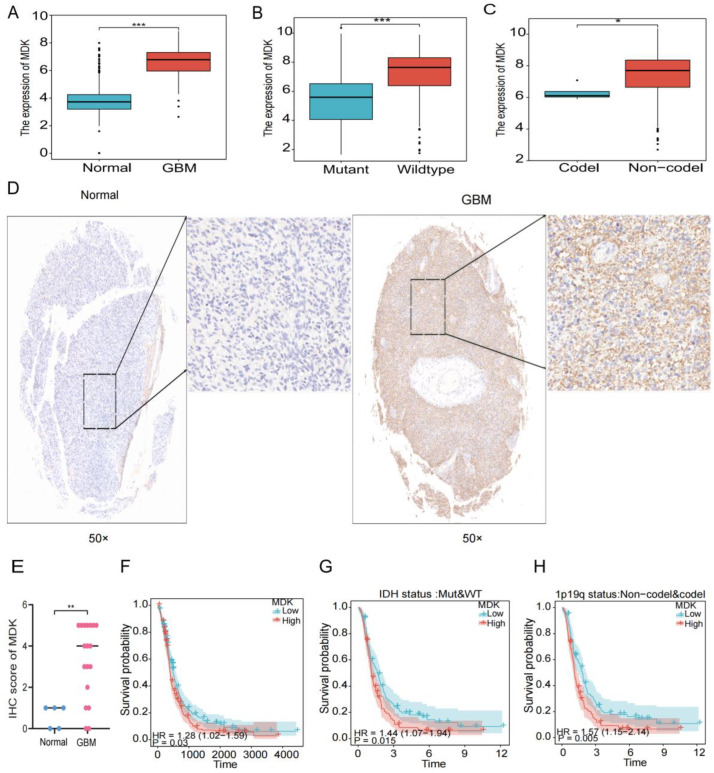
*MDK* is elevated in GBM patients and related to poor prognosis. (**A**) The gene expression of *MDK* in GBM in the TCGA database. (**B**,**C**) The gene expression of *MDK* in GBM with IDH status and 1p19q status in the TCGA database. (**D**,**E**) Representative IHC images of clinical samples, quantitative statistical analysis is shown below. Images on the right represent areas within the square magnified. (**F**–**H**) The K-M survival analysis of *MDK* in GBM in the TCGA database. (* *p* < 0.05, ** *p* < 0.01, *** *p* < 0.001, *p* < 0.05 was considered significant).

**Figure 12 biomedicines-12-00092-f012:**
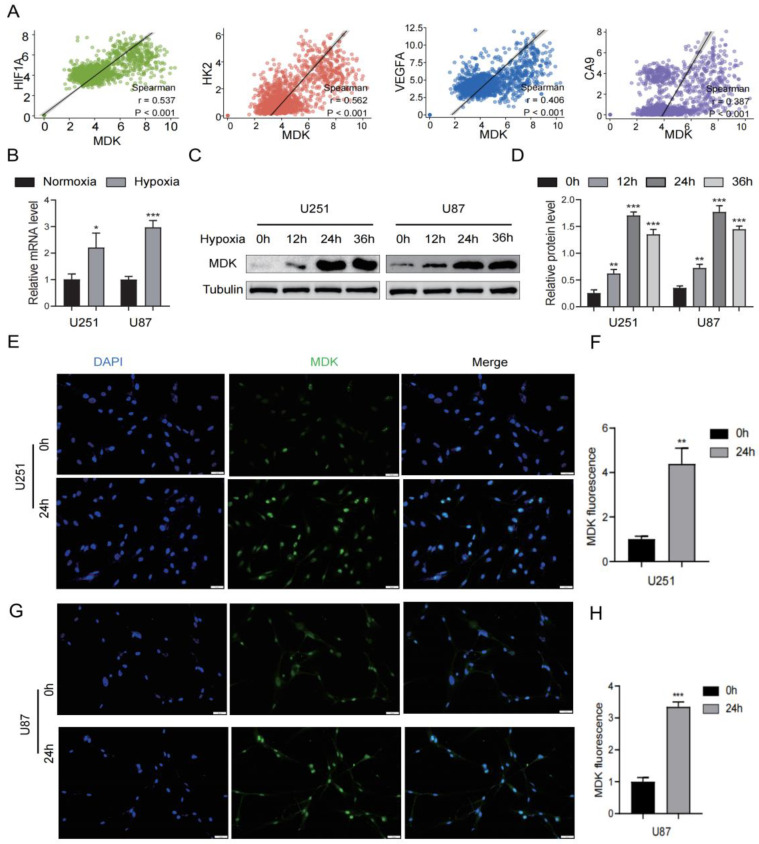
Hypoxia increases *MDK* expression in GBM cells. (**A**) Correlation between *MDK* and hypoxia-related markers such as HIF1F, HK2, VEGFA, and CA9 in the TCGA database. (**B**) The mRNA levels of *MDK* after GBM cells were cultured in hypoxia. (**C**,**D**) The protein levels of *MDK* after GBM cells were cultured in hypoxia, quantitative statistical analysis is shown on the right. (**E**–**H**) After GBM cells were cultured in hypoxia for 24 h, immunofluorescence was used to assess the *MDK* expression and relative fluorescence intensity was statistically analyzed. Scale bars: 20 μm. (* *p* < 0.05, ** *p* < 0.01, *** *p* < 0.001, *p* < 0.05 was considered significant).

**Figure 13 biomedicines-12-00092-f013:**
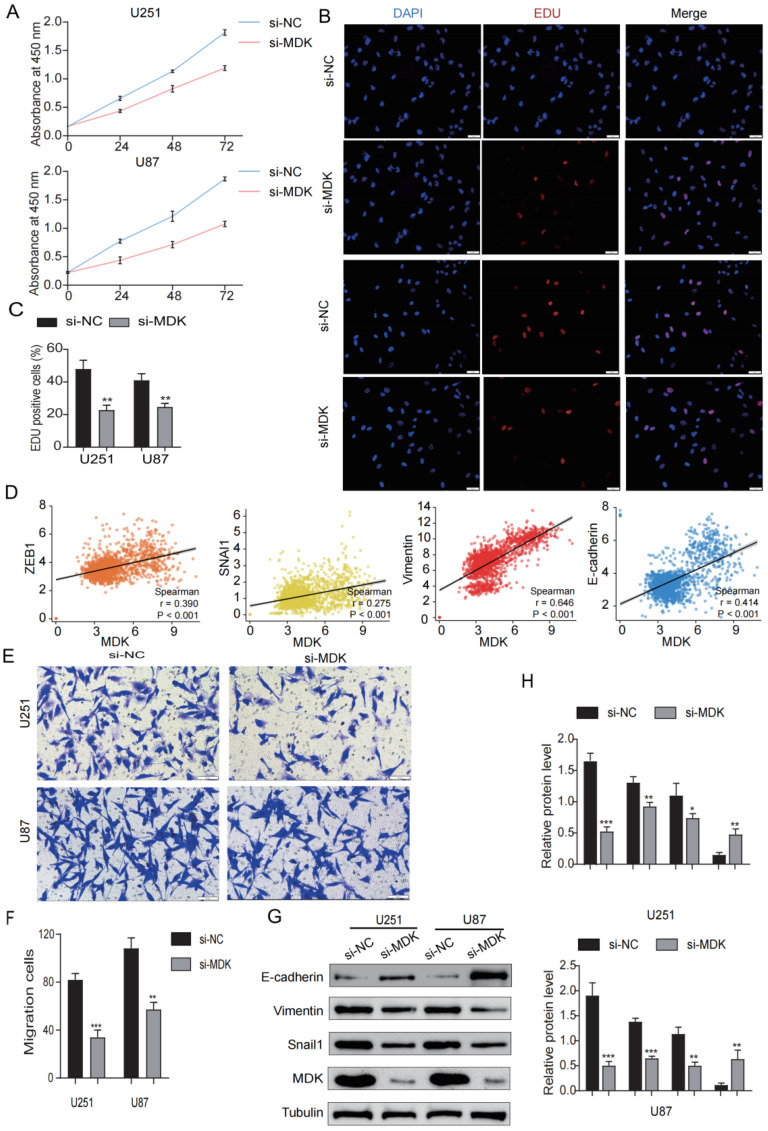
Knockdown *MDK* inhibits GBM cells’ proliferation and EMT. (**A**) CCK8 assay was utilized to detect the viability of GBM cells transfected with *MDK* plasmid. (**B**,**C**) EdU assay was utilized to detect the proliferation of GBM cells transfected with *MDK* plasmid, EdU-positive cells were defined as EdU cells/blue cells. (**D**) Correlation between *MDK* and EMT-related markers such as ZEB1, Snail1, Vimentin, and N-cadherin in the TCGA database. (**E**,**F**) The transwell assay was utilized to detect the invasive ability of GBM cells transfected with *MDK* plasmid and the results were statistically analyzed. Scale bars: 50 μm. (**G**,**H**) WB showing the protein levels of E-cadherin, Vimentin, Snail1, and *MDK* and quantitative statistical analysis is shown on the right. (* *p* < 0.05, ** *p* < 0.01, *** *p* < 0.001, *p* < 0.05 was considered significant).

**Figure 14 biomedicines-12-00092-f014:**
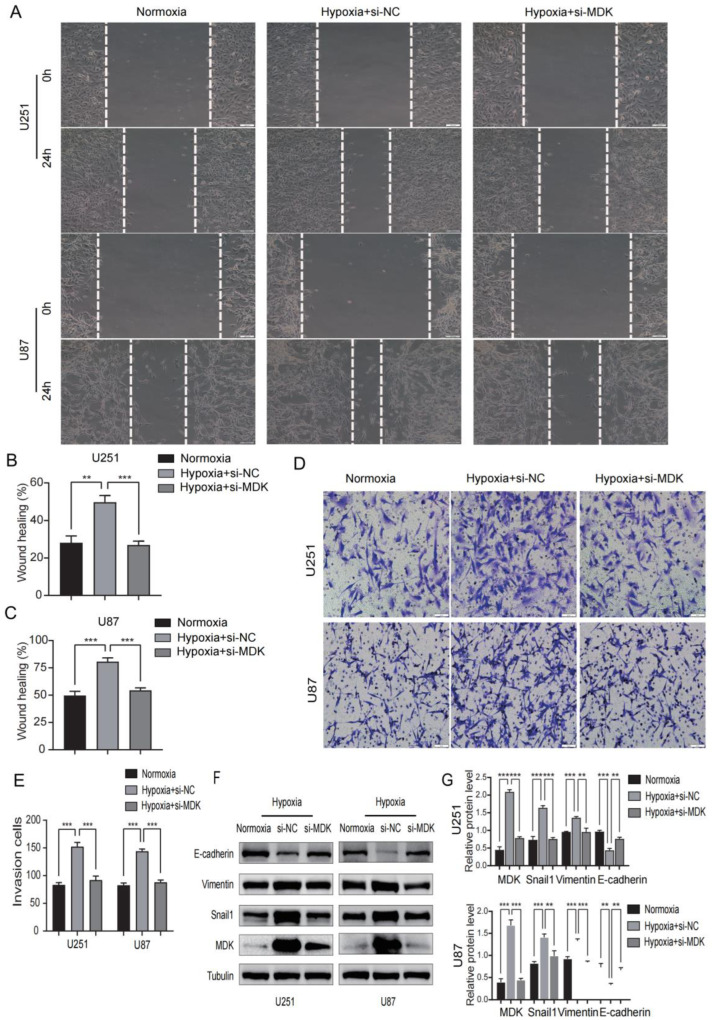
Knockdown *MDK* reverses hypoxia-induced EMT in GBM cells. (**A**–**C**) The wound healing assay was utilized to detect the migratory ability of MDA knockdown GBM cells in hypoxia and the results were statistically analyzed. Scale bars: 100 μm. (**D**,**E**) The transwell assay was utilized to detect the invasive ability of MDA knockdown GBM cells in hypoxia and quantitative statistical analysis is shown. Scale bars: 50 μm. (**F**,**G**) WB was performed to measure the expression of E-cadherin, Vimentin, Snail1, and *MDK*. Tubulin was utilized as internal control. ** *p* < 0.01, *** *p* < 0.001. Protein levels of E-cadherin, Vimentin, Snail1, and *MDK* and quantitative statistical analysis are shown on the right.

## Data Availability

https://www.jianguoyun.com/p/DdCiP5UQiKCaCxj23vQEIAA.

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
