# Peer review of "Identification of MDK as a Hypoxia- and Epithelial–Mesenchymal Transition-Related Gene Biomarker of Glioblastoma Based on a Novel Risk Model and In Vitro Experiments"

_biomedicines, 2024, doi:10.3390/biomedicines12010092_

Round 1

Reviewer 1 Report

Comments and Suggestions for Authors

In the study Gao et colleagues they used weighted gene co-expression network and DEGs analyses to processes the expression data of GBM in TCGA and GTEx databases. Based on LASSO and Cox analyses, they established a relevant risk model consisting of the MDK and STC1 genes, and subjected this model to the analyses of clinical, immune, mutation, drug treatment and functional enrichment. They found that the differences between high- and low-risk groups were manifested in tumor-related pathways, including PI3K-AKT and JAK-STAT. They selected MDK for in vitro experiments and proved that the hypoxia environment promoted the expression of MDK, and MDK knockdown reduced the proliferation, migration, and EMT of GBM cells induced by hypoxia.

The study is well done and can be published with a minor revision.

Minor points:

1.  Line 28. “In vivo experiment” should be replaced with “in vitro experiment”.

2.  Line 70. It is not clear why only two genes (STC1 and MDK) were selected from the gene risk panel presented on Fig 3A for risk model. The choice should be explained more logically.

3. The article contains virtually no data about the STC1 gene. Why was its transfection/knockdown not tested in vitro along with MDK?

4. Section 2.9 The number of patients and slices stained should be indicated (here or in the Result section).

5. Fig 11. The enlarged fragments should be presented in higher resolution.

6. The MDK is well known to be associated with GMB progression. The previous data should be discussed in more detail in Discussion section.

Author Response

Dear reviewer, Thank you for reviewing our manuscript and for the constructive comments, which greatly helped us to improve the manuscript. We have heavily revised our research. The manuscript was carefully revised and the point-by-point response was listed below. We hope that your comments have been addressed accurately. The revised manuscript was marked with yellow color and the responses were presented in black text.

1.Line 28. “In vivo experiment” should be replaced with “in vitro experiment”.

Thanks very much for discovering this error.We have replaced “In vivo experiment” to “in vitro experiment” in line 28.

 2.Line 70. It is not clear why only two genes (STC1 and MDK) were selected from the gene risk panel presented on Fig 3A for risk model. The choice should be explained more logically.

Thanks very much for your question.We have modified the sentence of“Our results suggest that the development of a differential expression risk score based on 2 genes (STC1 and MDK) has potential value in predicting the prognosis and guiding the treatment of GBM patients” to “The results of our study propose that the prognostic and therapeutic guidance potential for patients with GBM lies in the individualized risk scores derived from the expression levels of 2 genes (STC1 and MDK)” in line 69.

3.The article contains virtually no data about the STC1 gene. Why was its transfection/knockdown not tested in vitro along with MDK?

 Thank you very much for your question. In constructing our risk model, we included the STC1 and MDK genes. Since the role of STC1 gene in GBM has been studied in previous literature, the effect of MDK on GBM in anoxic environment has not been thoroughly studied. Therefore, we specifically selected MDK for in vitro validation in the context of GBM.

 4.Section 2.9 The number of patients and slices stained should be indicated (here or in the Result section).

Thanks for raising this important issue. The number of patients and slices stained have been indicated in Section 2.9.

 5.Fig 11. The enlarged fragments should be presented in higher resolution.

Thank you for your advice. Fig 11 resolution has been increased.

6.The MDK is well known to be associated with GMB progression. The previous data should be discussed in more detail in Discussion section.

Thanks for your suggestion.We have added the content in the second paragraph of the discussion.

Thank you again for your valuable advice and we look forward to learning more from you. If any further action is required, please let us know immediately. We look forward to hearing from you.

Reviewer 2 Report

Comments and Suggestions for Authors

Glioblastoma (GB) is the most common type of primary malignant brain tumour and is known for its aggressive nature. Unfortunately, tumour recurrence is frequent and there is a lack of effective treatments that improve clinical outcomes. Therefore, any new studies that could suggest new targets to treat the disease or improve clinical outcomes are important in advancing our understanding of GB. The manuscript in question is well-written, and the in-silico studies are well-designed and presented. Additionally, the in vitro experiments allow the researchers to confirm their hypothesis and in silico results. However, there are some issues that require review.

First of all, the TCGA and CGGA are different datasets with different populations (one of them is only with Chinese cohorts, which might be different to other races). Have the authors taken it into account? How have the authors aborted this topic? Are there other references that use CGGA as a validation cohort and TCGA as a study cohort? There are some statistical studies to confirm the homogeneity between both databases?

In the methodology section, concretely in 2.5, there are multiple immune cell algorithms, such as XCELL, TIMER, etc. I suggest adding the references for each other.

In relation to the results section, the flowchart in the study appears to be unclear. The graphic indicates that the validation cohort CGGA was used before the training cohort of TCGA. Furthermore, the figure specifies that only five patients exist in the TCGA cohort. According to the figure, there are 83 patients in the high-risk group and 83 in the low-risk group, but it seems like there are more patients in the dataset. Therefore, I suggest improving the explanation of section 3.1 and modifying the figure to make it more comprehensible.

Finally, why do authors select STC and MDK? There are other genes in the in-silico analysis. Are the authors performed more studies with the other genes? What criteria have you used to select only these two?

Author Response

Dear reviewer,Thank you so much for handling the review of our manuscript. We are sending you our revised manuscript entitled “Identification of MDK as a hypoxia and EMT-related gene biomarker of glioblastoma based on a novel risk model and in vitro experiments  ”. We appreciate the promoting comments to our study, and we have accepted and revised as recommended in this revised manuscript. The revised manuscript was marked with yellow color and the responses were presented in black text. We hope that the revision is acceptable, and your favorable consideration of our manuscript is greatly appreciated.Here are the point-by-point responses for your comments.

1.First of all, the TCGA and CGGA are different datasets with different populations (one of them is only with Chinese cohorts, which might be different to other races). Have the authors taken it into account? How have the authors aborted this topic?

Thanks for your questions.It is true that TCGA and CGGA are different database for different populations.However, we are compelled to acknowledge the limited of the brain glioma databases.Currently, the database for GBM is exclusively accessible through the TCGA, CGGA, and GTEx databases.As the times progress, I believe that an increasing number of databases encompassing diverse racial populations will become available, affording opportunities for their inclusion in our future research endeavors.

2.Are there other references that use CGGA as a validation cohort and TCGA as a study cohort?Thanks for your questions. There are other references that use CGGA as the validation cohort and TCGA as the research cohort1-3.

1 Wan, H. T., Su, Z. J., Guo, Z. S., Wen, P. & Hong, X. Y. Optimized risk stratification strategy for glioma patients based on the feature genes of poor immune cell infiltration patterns. J Cancer Res Clin Oncol 149, 13855-13874, doi:10.1007/s00432-023-05209-9 (2023).

2 Zeng, H. L., Li, H., Yang, Q. & Li, C. X. Transcriptomic Characterization of Copper-Binding Proteins for Predicting Prognosis in Glioma. Brain Sci 13, doi:10.3390/brainsci13101460 (2023).

3 Han, X. et al. IDH1(R132H) mutation increases radiotherapy efficacy and a 4-gene radiotherapy-related signature of WHO grade 4 gliomas. Sci Rep 13, 19659, doi:10.1038/s41598-023-46335-1 (2023).

3.There are some statistical studies to confirm the homogeneity between both databases?

Thanks for raising this important issue.We used the R package "limma" of the "normalizeBetweenArrays" function to reduce batch effects that may exist between or within the TCGA and CGGA cohorts which was also used and recognized in many articles1-3 4-6.

1 Lin, C. Y. et al. Membrane protein-regulated networks across human cancers. Nat Commun 10, 3131, doi:10.1038/s41467-019-10920-8 (2019).

2 Ritchie, M. E. et al. limma powers differential expression analyses for RNA-sequencing and microarray studies. Nucleic Acids Res 43, e47, doi:10.1093/nar/gkv007 (2015).

3 Zhang, M. et al. Integrative Analysis of DNA Methylation and Gene Expression to Determine Specific Diagnostic Biomarkers and Prognostic Biomarkers of Breast Cancer. Front Cell Dev Biol 8, 529386, doi:10.3389/fcell.2020.529386 (2020).

4.In the methodology section, concretely in 2.5, there are multiple immune cell algorithms, such as XCELL, TIMER, etc. I suggest adding the references for each other.

Thanks for your suggestion. Done as requested.

5.In relation to the results section, the flowchart in the study appears to be unclear. The graphic indicates that the validation cohort CGGA was used before the training cohort of TCGA. Furthermore, the figure specifies that only five patients exist in the TCGA cohort. According to the figure, there are 83 patients in the high-risk group and 83 in the low-risk group, but it seems like there are more patients in the dataset. Therefore, I suggest improving the explanation of section 3.1 and modifying the figure to make it more comprehensible.

Thanks for raising this important issue. Modifications have been implemented to the Section 3.1.

6.Finally, why do authors select STC and MDK? There are other genes in the in-silico analysis. Are the authors performed more studies with the other genes? What criteria have you used to select only these two?

Thank you for your question. These two genes form our risk model, which is derived from univariate cox analysis, LASSO regression analysis, and multivariate cox regression analyses. Because other genes were not involved in the construction of the risk model in this study, and our subsequent clinical and immune studies focused on the risk model, further studies on the remaining genes were not pursued. Our criteria were: univariate and multivariate cox analyses (p<0.05),LASSO regression analysis to remove overfitting to obtain the final gene.

Thank you again for your kind comments. If any further action is required, please let us know immediately. We look forward to hearing from you.

Reviewer 3 Report

Comments and Suggestions for Authors

Dear corresponding author:

After a through review of your bioinformatics focused article aiming at  designing detailed risk model to develop survival and prognosis predictions in glioblastoma supported. These findings are supported by in vitro assessments to determine MDK's mediatory role in EMT of tumor cells in hypoxic environment. 

The article is well organized and would be a good addition to special addition on method development.

Best regards, Ammar

Author Response

Dear reviewer,

Thank you for your acknowledgment and appreciation of the manuscript. I remain committed to furthering relevant research endeavors and wish you a joyful and fulfilling life.

Best regards,

Minqi Xia.

Reviewer 4 Report

Comments and Suggestions for Authors

Job well-done, the authors present a manuscripts that balances use of bioinformatics tool for target discovery with empirical validation of the findings. Overall

Comments:

U251 and U87 GBM cells are outdated cell models and it is unclear what the translational relevance of these findings will be.

Only one siRNA was used for the knock-down studies

The authors metions "in vivo experiments" in their abstract but these were not performed. What the manuscript includes is query of patient specimens. This needs to be clarified.

Although it is mentioned in the manuscript, it is unclear how the authors envision the use of the 2-gene prognostic signature being used in the clinic. Is this going to be used to inform treatment, predict outcome, early detection, will be included as part of the staging protocol for tumors? How do these findings improve or compliment what is already standard practice? Please expand on this in the discussion.

Comments on the Quality of English Language

A few typos and grammatical errors identified. These should be easily resolved during proofreading. 

Author Response

Dear reviewer: Thank you for your careful reading of the manuscript and constructive remarks. We have taken the comments on board to improve and clarify the manuscript.The revised manuscript was marked with yellow color and the responses were presented in black text.

1.The authors metions "in vivo experiments" in their abstract but these were not performed. What the manuscript includes is query of patient specimens. This needs to be clarified.

Thanks for raising this important issue.This is a clerical error on our part. We conducted in vitro experiments which has been corrected in the article.

2.Although it is mentioned in the manuscript, it is unclear how the authors envision the use of the 2-gene prognostic signature being used in the clinic. Is this going to be used to inform treatment, predict outcome, early detection, will be included as part of the staging protocol for tumors? How do these findings improve or compliment what is already standard practice? Please expand on this in the discussion.

Thank you for your question. The risk model was calculated by the expression level of two genes(STC1 and MDK), and each patient was given a risk score that was used to classify the patient into a high-risk or low-risk group. Based on our current findings, the risk score and patient grouping enable an assessment of sensitivity to drug therapy and a prediction of patient survival rates. This model holds promise as a biological target, potentially integrated into tumor staging protocols.It is essential to note that our research is currently in the early stages of basic experimentation, and the refinement or supplementation of established practices necessitates the accumulation of additional clinical data, execution of prospective studies, and even the initiation of clinical trials. We will further improve these aspects in the future research.This has already been explained in the discussion section.

Thank you again for your valuable advice and we look forward to learning more from you. If any further action is required, please let us know immediately. We look forward to hearing from you.